# Manganese Mediates Its Antiviral Functions in a cGAS-STING Pathway Independent Manner

**DOI:** 10.3390/v15030646

**Published:** 2023-02-28

**Authors:** Shaohua Sun, Yulin Xu, Ming Qiu, Sen Jiang, Qi Cao, Jia Luo, Tangjie Zhang, Nanhua Chen, Wanglong Zheng, Francois Meurens, Zongping Liu, Jianzhong Zhu

**Affiliations:** 1College Veterinary Medicine, Yangzhou University, Yangzhou 225009, China; 2Joint International Research Laboratory of Agriculture and Agri-Product Safety, Yangzhou University, Yangzhou 225009, China; 3Comparative Medicine Research Institute, Yangzhou University, Yangzhou 225009, China; 4Jiangsu Co-Innovation Center for Prevention and Control of Important Animal Infectious Diseases and Zoonoses, Yangzhou University, Yangzhou 225009, China; 5Swine and Poultry Infectious Diseases Research Center, Faculty of Veterinary Medicine, University of Montreal, St. Hyacinthe, QC J2S 2M2, Canada; 6Department of Veterinary Microbiology and Immunology, Western College of Veterinary Medicine, University of Saskatchewan, Saskatoon, SK S7N 5E2, Canada

**Keywords:** manganese, cGAS-STING pathway, RNA virus, antiviral function, redundant mechanisms

## Abstract

The innate immune system is the first line of host defense sensing viral infection. Manganese (Mn) has recently been found to be involved in the activation of the innate immune DNA-sensing cGAS-STING pathway and subsequent anti-DNA virus function. However, it is still unclear whether Mn^2+^ mediates host defense against RNA viruses. In this study, we demonstrate that Mn^2+^ exhibited antiviral effects against various animal and human viruses, including RNA viruses such as PRRSVs and VSV, as well as DNA viruses such as HSV1, in a dose-dependent manner. Moreover, cGAS and STING were both investigated in the Mn^2+^ mediated antiviral roles using the knockout cells made by the CRISPR-Cas9 approach. Unexpectedly, the results revealed that neither cGAS knockout nor STING knockout had any effect on Mn^2+^-mediated antiviral functions. Nevertheless, we verified that Mn^2+^ promoted the activation of the cGAS-STING signaling pathway. These findings suggest that Mn^2+^ has broad-spectrum antiviral activities in a cGAS-STING pathway independent manner. This study also provides significant insights into redundant mechanisms participating in the Mn^2+^ antiviral functions, and also indicates a new target for Mn^2+^ antiviral therapeutics.

## 1. Introduction

The innate immune system is the first line of host defense against multiple danger signals from pathogens or cellular damage. It encodes various host germline-encoded pattern recognition receptors (PRRs), which recognize pathogen-associated molecular patterns (PAMPs) and damage associated molecular patterns (DAMPs) [1,2]. To fight infection and achieve homeostasis, PRRs cause the activation of gene transcription or protease-dependent cytokine maturation, producing antiviral interferons (IFNs), proinflammatory cytokines, and chemokines [1,2]. Host antiviral innate immune responses are elicited during viral infections, and, upon engagement by viral infections, cells use different PRRs to sense viral nucleic acids [3]. The DNA-sensing PRRs are Toll-like receptor 9 (TLR9), cyclic GMP-AMP synthase (cGAS), stimulator of interferon genes (STING), absent in melanoma 2 (AIM2), and interferon gamma-inducible 16 (IFI16) [4]. RNA-sensing PRRs include endosomal TLR3, TLR7, TLR8, cytosolic retinoic acid inducible gene-I (RIG-I), melanoma differentiation-associated gene 5 (MDA5), NOD-like receptor pyrin domain containing 3 (NLRP3), and nucleotide-binding and oligomerization domain containing 2 (NOD2) [5].

The cGAS-STING pathway has been identified as the important DNA-sensing machinery in innate immunity against pathogens [6,7]. cGAS senses the presence of non-self and self-DNA, and utilizes substrates ATP and GTP to catalyze the production of the second messenger cyclic GMP-AMP (2′3′-cGAMP), which then activates the signaling adaptor protein STING [6,7]. Activated STING recruits the downstream TANK-binding kinase 1 (TBK1), and TBK1 is auto-phosphorylated [6,8]. Then, the transcription factor IRF3, which is recruited by STING and phosphorylated by TBK1, translocates to the nucleus and induces antiviral type I IFNs and IFN-stimulated genes (ISGs) [6,8]. Another transcription factor, NF-κB, is also activated by STING-TBK1 signaling, and drives proinflammatory gene expressions [6,9].

Manganese (Mn) is required as an enzymatic cofactor in many physiologic processes, such as protein and energy metabolism, immune function, development, reproduction, neuronal regulation, and antioxidant defenses [10,11,12]. Several canonical signaling pathways have been reported to be Mn-responsive, including the ataxia telangiectasia mutated 2 (ATM), p53, phosphatidylinositol 3 kinase (PI3K), insulin, and insulin-like growth factor-1 (IGF-1) pathways [13,14,15]. However, attention has been recently paid to Mn in the regulation of the cGAS-STING pathway, which exerts a potent host defense against DNA viruses [16]. Mn^2+^ was shown to increase the sensitivity of cGAS to double-stranded DNA (dsDNA) and its enzymatic activity. It also facilitates STING activity by boosting cGAMP-STING binding affinity [16]. Further studies have revealed that Mn^2+^ directly activates cGAS to induce a noncanonical catalytic synthesis of 2′3′-cGAMP, through similar overall conformation to dsDNA-activated cGAS [17,18].

Interestingly, recent publications have suggested the relevance of the cGAS-STING pathway in the process of RNA virus infections [19,20,21,22]. However, it is still largely unclear as to whether Mn^2+^ plays a role in cGAS-STING pathway-mediated anti-RNA virus infections. Moreover, it has not been fully elucidated as to how Mn^2+^ mediates its antiviral functions. In this study, we found that Mn^2+^ exerts a broad antiviral function against various viruses, including some RNA and DNA viruses, which is independent of the cGAS-STING pathway.

## 2. Materials and Methods

### 2.1. Cells and Viruses

Marc-145 cells (Elabscience, CL-0566) and HEK293T cells (ATCC, CRL-3216) were cultured in DMEM (HyClone Laboratories, Logan, UT, USA) containing 10% fetal bovine serum (FBS) and 1% penicillin-streptomycin solution at 37 °C with 5% CO_2_. Porcine alveolar macrophages (3D4/21, ATCC CRL-2843) were grown in RPMI 1640 (Hyclone Laboratories, Logan, UT, USA), supplemented with 10% FBS with 1% penicillin–streptomycin solution, and maintained at 37 °C with 5% CO_2_ in a humidified incubator. The viruses, including a DNA virus Herpes Simplex Virus 1 (HSV1-GFP), as well as RNA viruses Vesicular Stomatitis Virus (VSV-GFP) and Highly Pathogenic Porcine Reproductive and Respiratory Syndrome Viruses (HP-PRRSV XJ17-5-GFP and HP-PRRSV vaccine strain JXA1-R-GFP), were used as previously reported [22,23,24].

### 2.2. Cell Treatments

The Marc-145 cells were pretreated with different concentrations of Mn^2+^ (MnCl_2_·4H_2_O) (Sigma-Aldrich, St. Louis, MO, USA) (100–1000 μM) for 24 h, washed, and then infected with 0.1 multiplicity of infection (MOI) PRRSV XJ17-5 or JXA1-R. The 3D4/21 cells were also pretreated with Mn^2+^ (MnCl_2_·4H_2_O) (50–800 μM) for 24 h, washed and subsequently infected with 0.01 MOI HSV-1 or 0.001 MOI VSV. Additionally, 3D4/21 cells were transfected with cGAS agonist poly dA:dT (InvivoGen, Hong Kong, China) or STING agonist 2′3′-cGAMP (InvivoGen, Hong Kong, China) by using the transfection reagent Lipofectamine 2000 (ThermoFisher Scientific, Shanghai, China), and then exposed to Mn^2+^ treatment for 24 h.

### 2.3. Western Blot Analysis

Proteins were extracted in a radioimmunoprecipitation assay (RIPA) lysis buffer, mixed with a 4 × loading buffer in a 3:1 ratio, and boiled at 100 °C for 5–10 min. The protein samples were separated on 10% SDS-PAGE gels, and then transferred to PVDF membranes. After blocking with 5% skim milk solution at room temperature (RT) for 60 min, membranes were incubated with individual primary antibodies at 4 °C overnight. The primary antibodies include anti-GFP (HT801-01, TransGen, Beijing, China), STING (19851-1-AP, ProteinTech, Wuhan, China), cGAS (sc-515777, Santa Cruz Biotechnology, Dallas, Texas, USA), IRF3 (11904S, CST, Boston, MA, USA), p-IRF3 (Ser396) (MA5-14947, ThermoFisher Scientific, Shanghai, China), TBK1 (3504S, CST, Boston, MA, USA), p-TBK1 (5483S, CST, Boston, MA, USA) and β-actin (5057, CST, Boston, MA, USA). Secondary antibody HRP-conjugated goat anti-mouse or rabbit IgG (TransGen Biotech, Beijing, China) was used to incubate the membranes for 60 min at RT. Signals were detected using enhanced chemiluminescence (ECL) substrate (Tanon, Shanghai, China), and images were visualized with an imaging system (Tanon, Shanghai, China).

### 2.4. Quantitative Reverse Transcription Polymerase Chain Reaction (qRT-PCR)

Total RNA was extracted using TRIpure reagent (Aidlab, Beijing, China). The cDNA was synthesized using HiScript^®^ 1st Strand cDNA Synthesis Kit (Vazyme, Nanjing, China). The target gene expressions were examined using ChamQ Universal SYBR qPCR Master Mix (Vazyme, Nanjing, China) on a StepOne Plus real-time PCR system (Applied Biosystems, Foster City, CA, USA). The qPCR program was 95 °C for 30 s, followed by 40 cycles of 95 °C for 5 s and 60 °C for 1 min. β-actin served as an internal reference control. The relative mRNA levels were calculated using the 2^−ΔΔCT^ method. For all the qPCR assays, an efficiency comprised between 90 and 110% was measured. The sequence of qPCR primers used in this study are listed in Appendix A.

### 2.5. CRISPR gRNA Design and Preparation of Knockout (KO) Cells

The CRISPR gRNAs targeting porcine cGAS and monkey cGAS were designed using the web tool from Benchling (www.benchling.com (accessed on 4 January 2023)). For each gene, two gRNAs were chosen according to the predicted high scores, which are shown in Appendix A. The recombinant pX458-gRNA plasmids were obtained when the annealed gRNA encoding DNA sequences were cloned into the *Bbs* I site of pX458-EGFP. Marc-145 cells or 3D4/21 cells were transfected with the corresponding recombinant pX458-gRNA plasmids using Lipofectamine 2000. At 24 h post transfection, the GFP positive cells were sorted by a FACS Aria SORP cell sorter (Becton Dickinson, Franklin lakes, NJ, USA) and cultured in 96-well plates by limiting dilution for monoclonal growth. The individual cell clones were screened by PCR using primers, as shown in Appendix A. Briefly, the genomic PCR products were cloned into T vectors with a pClone007 versatile simple vector kit (TsingKe Biological Technology, Beijing, China). Base substitution, insertion, and deletion (ins/del) mutations were analyzed after the sequencing of inserted fragments, and cGAS^-/-^ monkey Marc-145 cells and cGAS^-/-^ porcine macrophages (3D4/21) were each acquired (Appendix A). In addition, the STING^-/-^ Marc-145 cells and STING^-/-^ 3D4/21 cells were previously obtained, and have been used in our lab [22,25].

### 2.6. Virus Tissue Culture Infectious Dose 50 (TCID50) Titrations

Marc-145 cells or 3D4/21 cells were seeded into 96-well plates, and then infected with 10-fold serial dilutions of various virus samples (Marc-145 cells for PRRSVs and 3D4/21 cells for VSV and HSV1). Next, the infected cell supernatants were replaced with fresh DMEM or RPMI 1640 containing 2% FBS, and the cells were monitored for the GFP fluorescence and cytopathic effects (CPE) characterized by cell clumping and shrinkage in Marc-145 cells or 3D4/21 cells after infections for 1–5 days. Finally, the viral titers were expressed as TCID50 and calculated using the method of Reed–Muench.

### 2.7. Dual-Luciferase Reporter Promoter Assay

The 293T cells were seeded in 96-well plates, followed by transfection the next day. Cells were co-transfected with reporter plasmids, ISRE-Firefly luc (Fluc) or IFNβ-Fluc (10 ng/well) and *Renilla* luciferase (Rluc) reporters (0.2 ng/well), plus the indicated porcine cGAS and STING plasmids or vector control (10–30 ng/well) using Lipofectamine 2000. The total DNA per well was normalized with control vectors to 50 ng. Twenty-four hours post transfection, the cells were treated with different concentrations of Mn^2+^ for another 24 h. Then, the cells were harvested, and luciferase activities were detected with the TransDetect Double-Luciferase Reporter Assay Kit (Vazyme, Nanjing, Jiangsu, China). The fold changes were calculated relative to control samples after Fluc normalization by the corresponding Rluc.

### 2.8. Statistical Analysis

The results were analyzed using the software GraphPad Prism v.6.0 and expressed as the mean ± standard deviation (SD). Statistical analysis was conducted by one way ANOVA, followed by Tukey’s post hoc test. The normality of the data distribution was assessed using the Shapiro–Wilk test. A *p* value of less than 0.05 was considered statistically significant.

## 3. Results

### 3.1. Mn^2+^ Exerted Antiviral Functions against PRRSV, VSV and HSV-1

To investigate the effect of Mn^2+^ on the different viruses, the Marc-145 cells and 3D4/21 cells were pretreated with various concentrations of Mn^2+^, and then the Marc-145 cells were infected with two PRRSV strains, whereas the 3D4/21 cells were infected with VSV and HSV-1, as shown in Figure 1A,B and Appendix A, Mn^2+^ (100, 200, 500, 800, and 1000 μM) inhibited PRRSV XJ17-5 replications in Marc-145 cells in a dose-dependent manner at both 24 h and 48 h post infection, as evidenced by Western blotting, GFP fluorescence, and TCID50 assay. Similarly, the Mn^2+^ treatment showed a comparable anti-PRRSV JXA1-R-GFP effect (Figure 1C,D and Appendix A). The VSV replication in 3D4/21 cells in the presence of Mn^2+^ (50, 100, 200, 500, and 800 μM) was examined, and the results revealed that Mn^2+^ suppressed VSV replication in a dose dependent manner in Western blotting, fluorescence microscopy, and TCID50 assay (Figure 1E,F and Appendix A). Furthermore, HSV-1 replication in the 3D4/21 cells was also decreased in a dose-dependent manner after the pretreatment with Mn^2+^ (Figure 1G,H and Appendix A). The cell viability of the Mn^2+^-treated cells was examined, and the results showed that Mn^2+^ up to 500 μM in Marc-145 cells and Mn^2+^ up to 200 μM in 3D4/21 cells did not exhibit any cytotoxicity (Appendix A).

### 3.2. Mn^2+^ Triggered Antiviral Activity against PRRSVs Was cGAS-STING Independent

Mn^2+^ has been found to promote the sensitivity of the cGAS-STING pathway for double-stranded DNA [16]. Thus, we first explored whether Mn^2+^ affects PRRSV replications depending on cGAS and STING. The cGAS^-/-^ Marc-145 cells (Appendix A) and STING^-/-^ Marc-145 cells were both utilized for PRRSV infections in the presence of Mn^2+^ (200 and 500 μM), and the anti-PRRSV activity by Mn^2+^ was examined. Western blotting and TCID50 assay showed that PRRSV XJ17-5 replication was upregulated in the cGAS^-/-^ Marc-145 cells relative to those in normal Marc-145 cells. However, Mn^2+^-triggered anti-PRRSV XJ17-5 activity did not appear altered in cGAS^-/-^ Marc-145 cells at both 24 h and 48 h post infection (Figure 2A,B). Similar results were obtained with the infections of PRRSV JXA1-R (Figure 2C,D).

Then, the effect of Mn^2+^ on PRRSV replication was examined in the STING^-/-^ Marc-145 cells. Western blotting and TCID50 assay showed that the PRRSV XJ17-5 replication was upregulated in the STING^-/-^ Marc-145 cells relative to normal Marc-145 cells. However, Mn^2+^ (200 and 500 μM) triggered anti-PRRSV XJ17-5 activity, which was retained at both 24 h and 48 h post infection (Figure 3A,B). Similar results were obtained with the PRRSV JXA1-R infections (Figure 3C,D). Together, the results indicated that, although the cGAS-STING pathway plays a role in the anti-PRRSV innate immunity, it is not required for Mn^2+^-triggered anti-PRRSV activity.

### 3.3. Mn^2+^ Triggered Antiviral Activity against VSV and HSV-1 Was Independent of cGAS-STING

To detect whether the antiviral functions of Mn^2+^ against VSV and HSV-1 in 3D4/21 cells were dependent on cGAS and STING, the cGAS^-/-^ 3D4/21 cells (Appendix A) and STING^-/-^ 3D4/21 cells were both examined after virus infections. The results showed that the replications of VSV and HSV-1 were both upregulated in cGAS^-/-^ 3D4/21 cells, compared with those in normal 3D4/21 cells. However, the antiviral functions of Mn^2+^ (100 and 200 μM) were not attenuated in cGAS^-/-^ 3D4/21 cells, as evidenced by Western blotting and TCID50 assay (Figure 4A–D). The VSV and HSV-1 replications were also upregulated in STING^-/-^ 3D4/21 cells compared with those in normal 3D4/21 cells, but the Mn^2+^ (100 and 200 μM)-triggered antiviral activities were not diminished in STING^-/-^ 3D4/21 cells (Figure 5A–D). Altogether, the results clearly suggest that the cGAS-STING pathway is not essential for Mn^2+^ triggered antiviral functions, despite the important role that the cGAS-STING pathway plays in innate antiviral immunity.

### 3.4. Mn^2+^ Treatment Promoted cGAS-STING Signaling Activity

To further validate whether Mn^2+^ can activate the cGAS-STING pathway, the effect of Mn^2+^ on the cGAS-STING signaling pathway was investigated in both 293T and 3D4/21 cells. In 293T cells, the co-transfection of porcine cGAS and STING activated downstream IFNβ and ISRE promoter activities, as well as IFNβ and ISG60 gene transcriptions (Figure 6A,B), whereas the Mn^2+^ (100 and 200 μM) treatments increased the two promoter activities and IFNβ and ISG60 transcriptions in dose-dependent manners (Figure 6A,B). In the 3D4/21 cells, both cGAS agonist polydA:dT and STING agonist 2′3′-cGAMP activated downstream IFNβ and ISG15 gene transcriptions, and the phosphorylations of TBK and IRF3 in Western blotting (Figure 6C,D). Similarly, Mn^2+^ (100 and 200 μM) further promoted the IFNβ and ISG15 transcriptions, and the phosphorylations of TBK and IRF3 in dose-dependent manners (Figure 6C,D). The results suggested that the cGAS-STING activation may be one of the redundant mechanisms of Mn^2+^-triggered antiviral functions in cells.

## 4. Discussion

The element Mn is critical for almost all forms of life [10,11,12]. Cytosolic Mn^2+^ has been reported to be involved in the dsDNA-sensing activity of cGAS, and protects against DNA viruses [16]. Recent studies have shown that Mn^2+^ could directly activate cGAS, which is independent of dsDNA [17,18]. In addition, the overlapping mechanisms between the antiviral innate immunity developed against RNA and DNA viruses have been reviewed previously [26]. Many RNA viruses of families *Flaviviridae*, *Coronaviridae*, and *Arteriviridae* have been found to be associated with the cGAS-STING pathway [19,20,21,22,27]. Likewise, in our study, we demonstrated that Mn^2+^ exerts antiviral functions against a DNA virus (HSV-1) and some RNA viruses (PRRSV XJ17-5, PRRSV JXA1-R, and VSV) in a dose-dependent manner. Mn^2+^ exhibited a broad antiviral activity, which is similar to that mediated by the cGAS-STING pathway. At the first glance, it is logical to deduce that Mn^2+^ exerts antiviral activity by acting on the cGAS-STING signaling pathway. However, further investigation revealed that Mn^2+^-triggered antiviral activity is cGAS-STING independent, suggesting that there is another cell mechanism which mediates the Mn^2+^ antiviral functions.

Previously, Mn^2+^ has been found to participate in the phosphorylation of p53 [15,28]. p53 is a tumor suppressor gene, and functions most commonly in cell cycle arrest, differentiation, and apoptosis [29]. Moreover, p53 has been reported to be involved the regulation of antiviral functions [30,31,32]. For example, p53 overexpression represses HIV-1 long terminal repeat (LTR) transcriptional elongation by preventing the phosphorylation of serine 2 of the pol II C-terminal domain (CTD), resulting in the inhibition of HIV-1 transcription and replication [30]. Influenza virus infection promotes the activation of the p53 pathway leading to apoptosis, and suppression of p53 activity contributes to influenza virus infection [31]. Similarly, depletion of p53 was shown to promote porcine epidemic diarrhea virus (PEDV) infection susceptibility [32]. p53 has robust antiviral immunity by activation of the IFN pathway and the induction of several antiviral proteins [33,34]. Miciak et al. have identified p53 as contributing to the perpetuation of IFN signaling through ISG-dependent positive feedback loops [35]. Hao et al. discovered that p53 facilitates IFN signaling and secretion, and activates ISREs and ISG expression during viral infection [32].

In addition, a recent report found that Mn^2+^ alone activates the phosphorylation of TBK1 with ATM involved, and enhances DNA- or RNA-mediated innate immune responses [36]. TBK1 is the downstream mediator of multiple DNA sensors, such as Ku70, IFI16, cGAS, and DDX41 [37]. Similarly, it is also involved in the signaling pathway of RNA sensors, such as RIG-I and MDA5 [38,39]. Here, our study demonstrates that Mn^2+^ has antiviral functions against RNA viruses (PRRSVs, VSV) and a DNA virus (HSV-1) in cGAS^-/-^ and STING^-/-^ cells. Mn^2+^ may initiate ATM-TBK1 signaling to exert its broad-spectrum antiviral functions [36]. Together, we speculate that Mn^2+^ may induce the activation of p53 and/or ATM-TBK1 to establish a powerfully antiviral state through cGAS-STING independent signaling pathways.

Nevertheless, we observed that the cGAS-STING signaling is indeed activated by Mn^2+^ treatment. This observation is consistent with the previous discovery that Mn^2+^ suppressed virus replicates by sensitizing both cGAS and STING [16], and suggests that cGAS-STING is one of the cell machineries that mediates Mn^2+^ antiviral functions. In our study, although Mn^2+^-mediated antiviral functions via a cGAS-STING independent pathway, it was also identified as promoting cGAS-STING signaling activity. In which case, what is the relationship between cGAS-STING signaling and other cell machinery triggered by Mn^2+^? These cell machineries must be redundant, and together participate in Mn^2+^-triggered antiviral functions. However, the exact molecular mechanisms underlying Mn^2+^-mediated antiviral functions merit further exploration for a complete elucidation.

In summary, we demonstrated that Mn^2+^ inhibits PRRSV XJ17-5, PRRSV JXA1-R, VSV, and HSV-1 replications in a cGAS-STING independent manner. Our results reveal that Mn^2+^ may harbor multiple cellular mechanisms to exert its broad-spectrum antiviral activity, and suggest that Mn^2+^ has the potential to be used not only as an antiviral therapeutic, but also as the immune adjuvant in some animal vaccines.

## Figures and Tables

**Figure 1 viruses-15-00646-f001:**
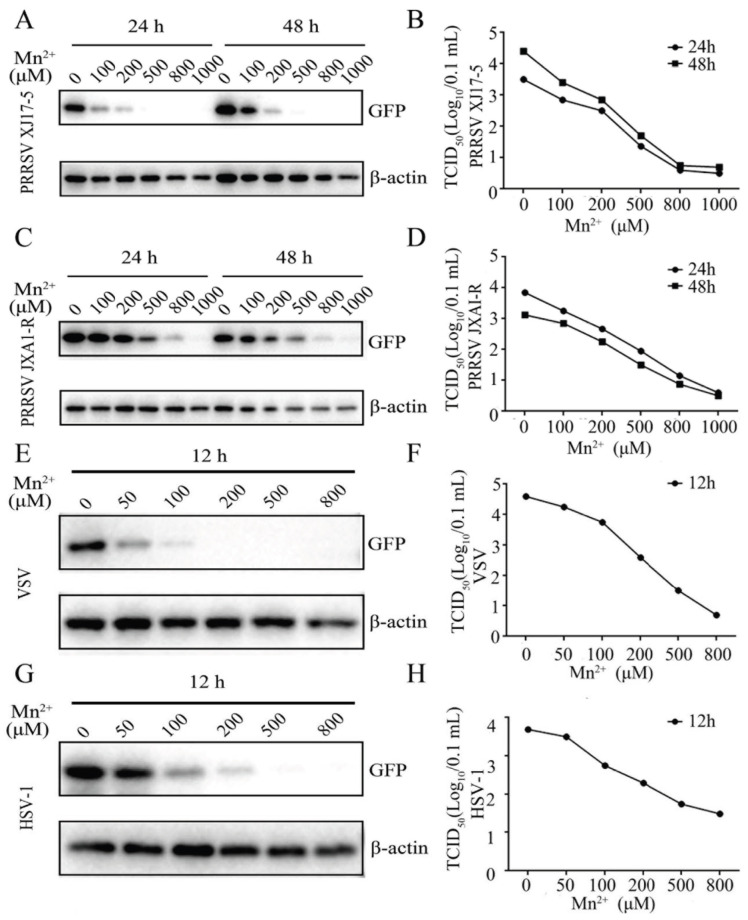
Mn^2+^ exerts antiviral functions against PRRSV, VSV, and HSV-1. (**A**–**D**) Marc-145 cells were pretreated with Mn^2+^ (0, 100, 200, 500, 800, and 1000 μM) for 24 h, and then infected with 0.1 MOI HP-PRRSV-GFP XJ17-5 (**A**,**B**) or HP-PRRSV-GFP JXA1-R (**C**,**D**) for 24 h and 48 h, respectively. (**E**–**H**) The 3D4/21 cells were pretreated with Mn^2+^ (0, 50, 100, 200, 500, and 800 μM) for 24 h, and then infected with 0.001 MOI VSV-GFP (**E**,**F**) or 0.01 MOI HSV-1-GFP (**G**,**H**) for 12 h. The GFP and β-actin protein levels were detected using Western blot analysis (**A,C,E,G**), and the virus titers in the supernatants were examined by TCID50 assay (**B,D,F,H**).

**Figure 2 viruses-15-00646-f002:**
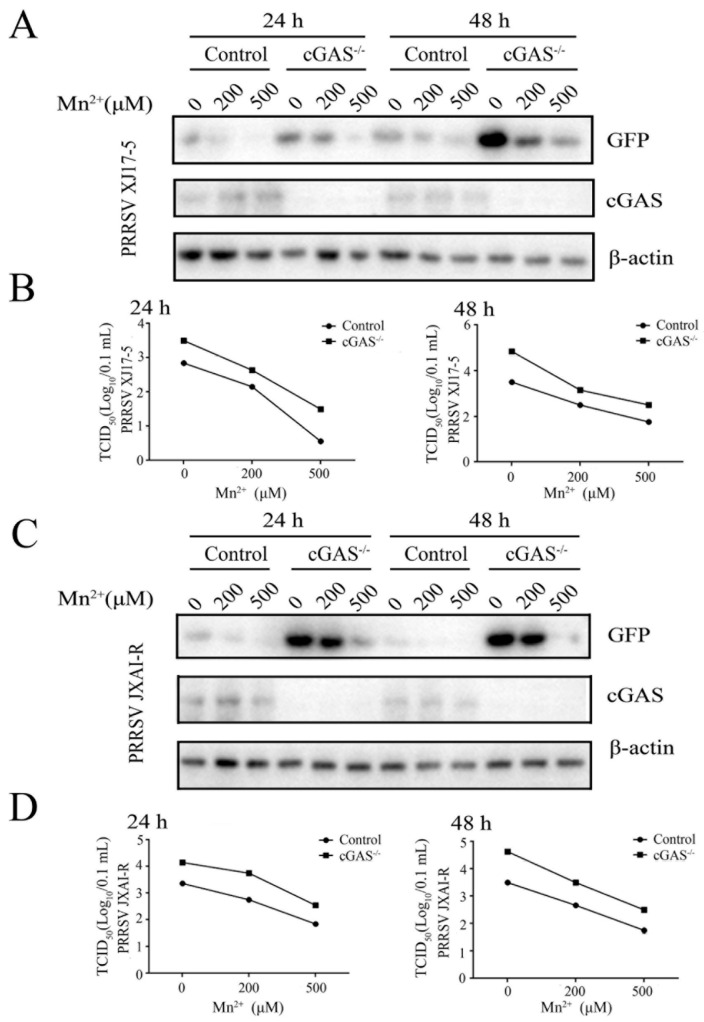
Mn^2+^ mediates antiviral functions against PRRSV in the cGAS-independent manner. The cGAS^-/-^ Marc-145 cells and normal Marc-145 control cells were pretreated with Mn^2+^ (0, 200, and 500 μM) for 24 h, and then infected with 0.1 MOI HP-PRRSV XJ17-5 (**A**,**B**) or 0.1 MOI PRRSV JXA1-R (**C**,**D**) for 24 h and 48 h, respectively. The cell samples were collected and subjected to Western blot analysis using anti-GFP, anti-cGAS, and anti-β-actin (**A**,**C**). The virus titers in the supernatants were detected using TCID50 assay (**B**,**D**).

**Figure 3 viruses-15-00646-f003:**
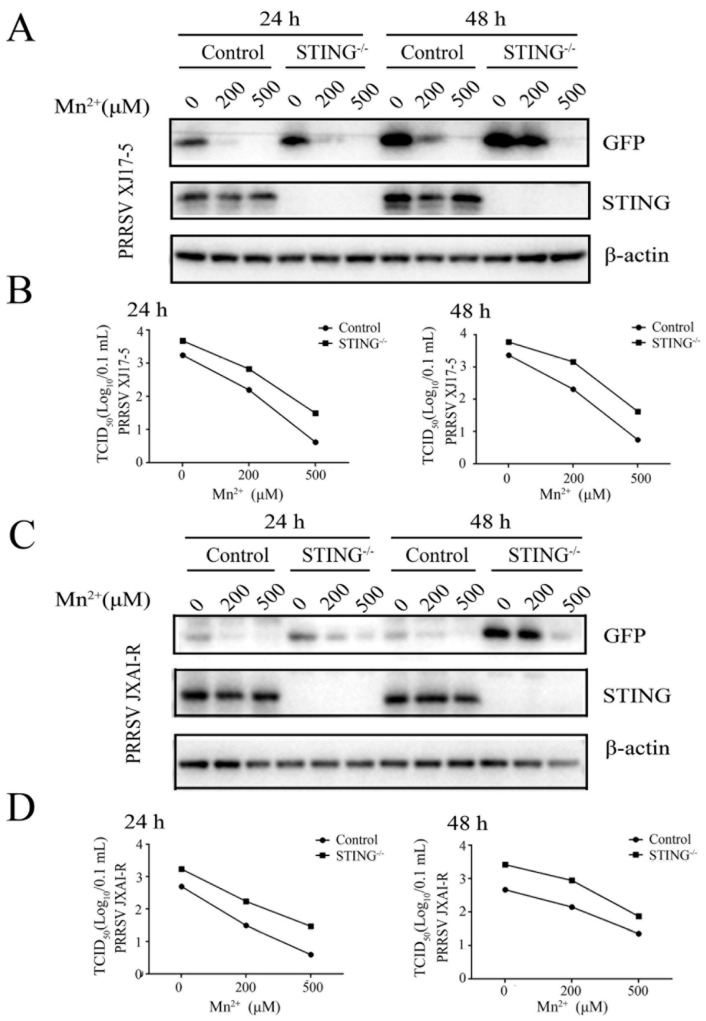
Mn^2+^ mediates antiviral functions against PRRSV in the STING-independent manner. The STING^-/-^ Marc-145 cells and normal Marc-145 control cells were pretreated with Mn^2+^ (0, 200, and 500 μM) for 24 h, and then infected with either 0.1 MOI HP-PRRSV XJ17-5 (**A**,**B**) or 0.1 MOI PRRSV JXA1-R (**C**,**D**) for 24 h and 48 h, respectively. The cell samples were collected and subjected to Western blot analysis using anti-GFP, anti-STING, and anti-β-actin (**A**,**C**). The virus titers in the supernatants were detected using TCID50 assay (**B**,**D**).

**Figure 4 viruses-15-00646-f004:**
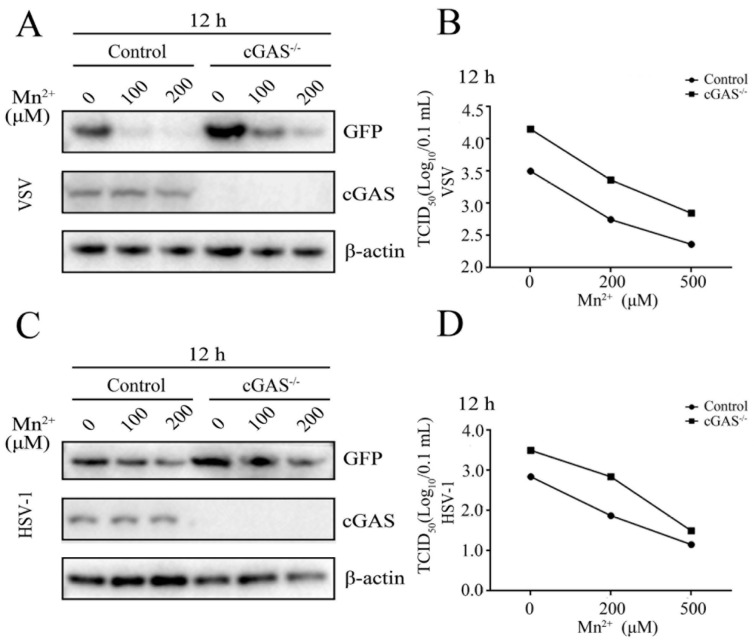
Mn^2+^ mediates antiviral functions against VSV and HSV-1 in the cGAS-independent manner. (**A**,**B**) cGAS^-/-^ 3D4/21 cells and normal 3D4/21 control cells were pretreated with Mn^2+^ (0, 100, and 200 μM) for 24 h, and then infected with 0.001 MOI VSV for 12 h. (**C**,**D**) cGAS^-/-^ 3D4/21 cells and normal 3D4/21 cells were pretreated with Mn^2+^ (0, 100, and 200 μM) for 24 h, and subsequently infected with 0.01 MOI HSV-1 for 12 h. The cell samples were collected and subjected to Western blot analysis using anti-GFP, anti-cGAS, and anti-β-actin (**A**,**C**). The virus titers in the supernatants were detected using TCID50 assay (**B**,**D**).

**Figure 5 viruses-15-00646-f005:**
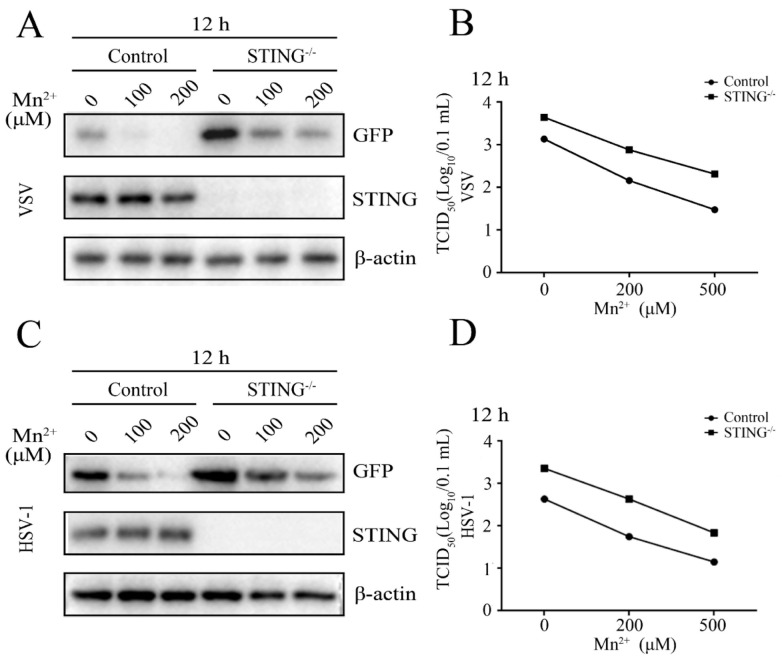
Mn^2+^ mediates antiviral functions against VSV and HSV-1 in the STING-independent manner. (**A**,**B**) STING^-/-^ 3D4/21 cells and normal 3D4/21 control cells were pretreated with Mn^2+^ (0, 100, and 200 μM) for 24 h, and then infected with 0.001 MOI VSV for 12 h. (**C**,**D**) STING^-/-^ 3D4/21 cells and normal 3D4/21 cells were pretreated with Mn^2+^ (0, 100, and 200 μM) for 24 h, and subsequently infected with 0.01 MOI HSV-1 for 12 h. The cell samples were collected and subjected to Western blot analysis using anti-GFP, anti-STING, and anti-β-actin (**A**,**C**). The virus titers in the supernatants were detected using TCID50 assay (**B**,**D**).

**Figure 6 viruses-15-00646-f006:**
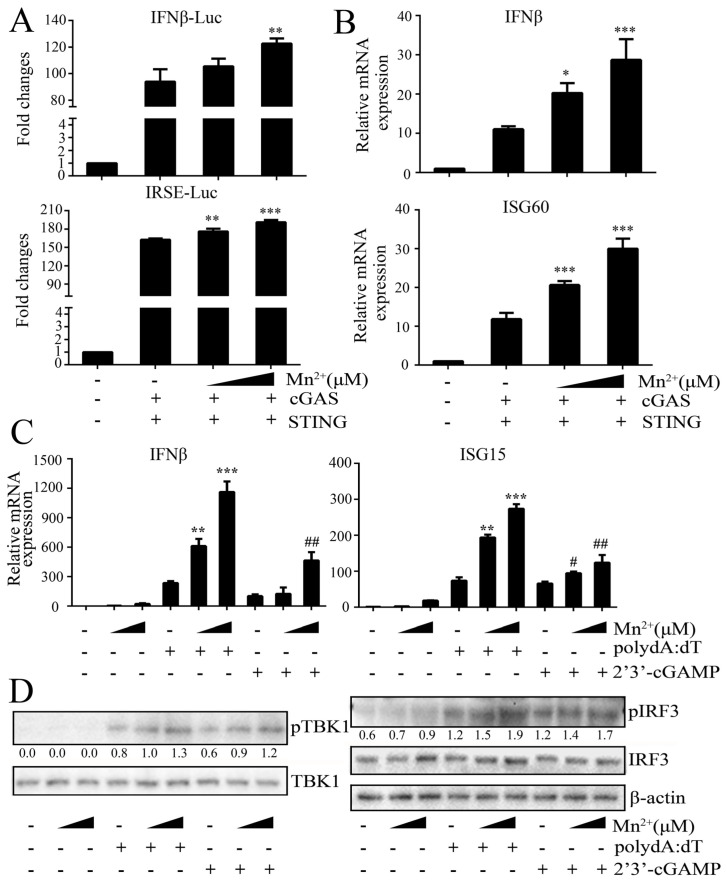
Mn^2+^ promotes cGAS-STING signaling activity. (**A**) HEK293T cells in 96-well plates were con-transfected with 20 ng cGAS-HA and 10 ng STING-GFP, plus 10 ng IFNβ-luc or ISRE-luc and 0.2 ng pRL-TK plasmid for 24 h. Cells were then treated with Mn^2+^ (0, 100, and 200 μM) for 24 h, and the luciferase activities were examined. (**B**) HEK293T cells in 24-well plates were co-transfected with 400 ng cGAS-HA and 400 ng STING-GFP plasmids for 24 h. Cells were then exposed to Mn^2+^ (0, 100, and 200 μM) for another 24 h, and subjected to RT-qPCR for downstream gene detection. (**C**,**D**) The 3D4/21 cells in 24-well plates were transfected with polydA:dT (1 μg/mL) or 2′3′-cGAMP (2 μg/mL) for 8  h using lipofectiamine 2000. Twenty-four hours post transfection, cells were incubated with Mn^2+^ (0, 100, and 200 μM) for another 24 h, and then downstream gene expressions were detected using RT-qPCR (**C**) and Western blot analysis with the indicated antibodies (**D**). The protein band density values of p-TBK1/TBK1 and p-IRF3/IRF3 were presented to indicate the levels of phosphorylations of TBK1 and IRF3. * or #, *p* < 0.05. ** or ##, *p* < 0.01. ***, *p* < 0.001 versus mock treatments.

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
