# Peer review of "Manganese Mediates Its Antiviral Functions in a cGAS-STING Pathway Independent Manner"

_viruses, 2023, doi:10.3390/v15030646_

Round 1
Reviewer 1 Report
Authors demonstrated that Mn2+ exhibits antiviral effects and inhibits RNA and DNA viruses replications in a cGAS-STING independent manner. Their results reveal that Mn2+ may harbor multiple cellular mechanisms to exert its broad spectrum antiviral activity. Moreover authors suggest that the Mn2+ has the potential to be used not only as an antiviral therapeutics but also as the immune adjuvant in some animal vaccines.
The paper is very interesting, but I see a fundamental flaw in it that temporarily disqualifies the manuscript for publication. I did not find in the text, neither in the Materials and Methods section nor in the Results section, the concentration of manganese ions. The authors mentioned several times about "various concentrations of M2+" but do not quote their values.
Author Response
Authors demonstrated that Mn2+ exhibits antiviral effects and inhibits RNA and DNA viruses replications in a cGAS-STING independent manner. Their results reveal that Mn2+ may harbor multiple cellular mechanisms to exert its broad spectrum antiviral activity. Moreover authors suggest that the Mn2+ has the potential to be used not only as an antiviral therapeutics but also as the immune adjuvant in some animal vaccines.
The paper is very interesting, but I see a fundamental flaw in it that temporarily disqualifies the manuscript for publication. I did not find in the text, neither in the Materials and Methods section nor in the Results section, the concentration of manganese ions. The authors mentioned several times about "various concentrations of M2+" but do not quote their values.
Answer: Thanks for the appreciation and concern by the reviewer. We added the information of concentrations of manganese ions in the Materials and Methods section, as well as in the Results section. We also incorporated the cytotoxicity data of Manganese ions in both Marc-145 cells and 3D4/21 cells, as the Supplementary Fig 3 in the revised manuscript.
Reviewer 2 Report
Sun and colleagues investigated the role of Mn+ ions in the antiviral activity of the cGAS-STING pathway. While originally implicated in cytoplasmic DNA sensing, recently the cGAS-STING-mediated signaling was shown to be important for the protection against at least some RNA viruses. Also, it has been shown that Mn+ released from the intracellular organelles in response to viral infection directly potentiates the sensitivity of cGAS-STING signaling. The authors incubated different cell lines in the increasing concentration of Mn+ and showed that such pre-incubation inhibits infection of DNA (HSV) and RNA (PRRSV, VSV) viruses. They also confirmed that Mn+ treatment promotes the cGAS-STING-dependent expression of interferon and interferon-stimulated genes. Unexpectedly, when using cGAS and STING knockout cells, the antiviral effect of Mn+ treatment was not eliminated, indicating that Mn+ exerts antiviral function through different mechanisms. The data are interesting, but the manuscript should be improved:
Fig.1 It is important to provide data on the cytotoxicity of Mn+ treatment in the conditions used for the infection experiments. Otherwise, it cannot be excluded that the inhibition of replication may be non-specific in the cells treated with relatively high Mn+ concentrations for a prolonged period of time.
Fig. 6D the phosphorylation signals are not obviously different between control and Mn-treated cells. The data on the reproducibility of the results and the statistical significance are required.
Minor comments:
Line 48. “aggression” here is not necessary. Just “pathogens” will suffice
Line 173. Remove “on the other hand”.Tthe VSV results are in no way contradictory to the inhibition of other viruses
Lne 227 Remove “on the other hand”. The results are similar to those previously described for CAS-/- cells
Line 229. Remove “Again”.
Line 231. It is not clear from this sentence what plays the important role in the antiviral innate immunity, Mn+ or cGAS-STING pathway
Author Response
Sun and colleagues investigated the role of Mn+ ions in the antiviral activity of the cGAS-STING pathway. While originally implicated in cytoplasmic DNA sensing, recently the cGAS-STING-mediated signaling was shown to be important for the protection against at least some RNA viruses. Also, it has been shown that Mn+ released from the intracellular organelles in response to viral infection directly potentiates the sensitivity of cGAS-STING signaling. The authors incubated different cell lines in the increasing concentration of Mn+ and showed that such pre-incubation inhibits infection of DNA (HSV) and RNA (PRRSV, VSV) viruses. They also confirmed that Mn+ treatment promotes the cGAS-STING-dependent expression of interferon and interferon-stimulated genes. Unexpectedly, when using cGAS and STING knockout cells, the antiviral effect of Mn+ treatment was not eliminated, indicating that Mn+ exerts antiviral function through different mechanisms. The data are interesting, but the manuscript should be improved:
Thanks for the appreciation.
Fig.1 It is important to provide data on the cytotoxicity of Mn+ treatment in the conditions used for the infection experiments. Otherwise, it cannot be excluded that the inhibition of replication may be non-specific in the cells treated with relatively high Mn+ concentrations for a prolonged period of time.
Answer: Following the suggestion of the reviewer, we did the cell viability assay with Mn2+ treatment as in the Supplementary Fig 3. The results showed that Mn2+ up to 500 μM in Marc-145 cells and Mn2+ up to 200 μM in 3D4/21 cells did not exhibit any cytotoxicity.
Fig. 6D the phosphorylation signals are not obviously different between control and Mn-treated cells. The data on the reproducibility of the results and the statistical significance are required.
Answer: We quantified the p-TBK1/TBK1 and p-IRF3/IRF3 ratios of protein band density in Fig 6D. The levels of p-TBK1/TBK and p-IRF3/IRF3 indeed increase obviously upon Mn2+ treatments as shown by the quantified values. The updated Fig 6D is presented in the revised manuscript.
Minor comments:
Line 48. “aggression” here is not necessary. Just “pathogens” will suffice
Answer: corrected.
Line 173. Remove “on the other hand”.Tthe VSV results are in no way contradictory to the inhibition of other viruses
Answer: corrected.
Lne 227 Remove “on the other hand”. The results are similar to those previously described for CAS-/- cells
Answer: corrected.
Line 229. Remove “Again”.
Answer: Removed.
Line 231. It is not clear from this sentence what plays the important role in the antiviral innate immunity, Mn+ or cGAS-STING pathway
Answer: it should be cGAS-STING pathway, and it has been updated.